# Bioinformatic Analyses of Broad H3K79me2 Domains in Different Leukemia Cell Line Data Sets

**DOI:** 10.3390/cells11182830

**Published:** 2022-09-10

**Authors:** Prerna Sharma, Hedieh Sattarifard, Narges Fatemiyan, Ted M. Lakowski, James R. Davie

**Affiliations:** 1Department of Biochemistry and Medical Genetics, University of Manitoba, Winnipeg, MB R3E 0J9, Canada; 2College of Pharmacy Pharmaceutical Analysis Laboratory, University of Manitoba, Winnipeg, MB R3E 0V9, Canada; 3CancerCare Manitoba Research Institute, CancerCare Manitoba, Winnipeg, MB R3E 0V9, Canada

**Keywords:** leukemia, broad H3K79me2 domains, DOT1L, epigenetics, chromatin accessibility

## Abstract

A subset of expressed genes is associated with a broad H3K4me3 (histone H3 trimethylated at lysine 4) domain that extends throughout the gene body. Genes marked in this way in normal cells are involved in cell-identity and tumor-suppressor activities, whereas in cancer cells, genes driving the cancer phenotype (oncogenes) have this feature. Other histone modifications associated with expressed genes that display a broad domain have been less studied. Here, we identified genes with the broadest H3K79me2 (histone H3 dimethylated at lysine 79) domain in human leukemic cell lines representing different forms of leukemia. Taking a bioinformatic approach, we provide evidence that genes with the broadest H3K79me2 domain have known roles in leukemia (e.g., *JMJD1C*). In the mixed-lineage leukemia cell line MOLM-13, the *HOXA9* gene is in a 100 kb broad H3K79me2 domain with other *HOXA* protein-coding and oncogenic long non-coding RNA genes. The genes in this domain contribute to leukemia. This broad H3K79me2 domain has an unstable chromatin structure, as was evident by enhanced chromatin accessibility throughout. Together, we provide evidence that identification of genes with the broadest H3K79me2 domain will aid in generating a panel of genes in the diagnosis and therapeutic treatment of leukemia in the future.

## 1. Introduction

Leukemia, a blood-borne cancer, typically results from genetic and epigenetic events that deregulate gene expression. The four main types of leukemia include acute myeloid leukemia (AML), chronic myeloid leukemia (CML), acute lymphocytic leukemia (ALL), and chronic lymphocytic leukemia. Epigenetics studies molecules and mechanisms that modify gene-activity states in the context of the same DNA sequence [1]. Epigenetic modifiers of chromatin include players categorized as writers, readers, and erasers. Writers introduce chemical modifications on DNA and histones; readers are effector proteins containing, for example, methyl-lysine-binding motifs, which identify and interpret those chemical changes; erasers are enzymes that remove the chemical tags.

Mixed-lineage leukemia (MLL) is a form of leukemia occurring predominantly in children that is a consequence of a chromosomal rearrangement of 11q23 [2,3,4,5]. These rearrangements can fuse otherwise independent genes into a hybrid gene, resulting in fusion proteins containing domains initially encoded by separate genes. In MLL, the *KMT2A* gene, which codes for an H3K4 lysine methyltransferase [6], fuses with another protein-coding gene, generating an oncoprotein [7]. In this fusion, KMT2A retains its *N*-terminus (KMT2A^N^), which is placed in-frame with the *C*-terminus of one of several fusion partners, creating an oncogenic chimeric protein [7,8]. Hence, the KMT2A fusion protein is composed of KMT2A^N^ and a fusion partner. KMT2A^N^ does not contain the H3K4 methylation SET domain but it does contain a CXXC domain that is involved in binding to nonmethylated CpG-rich promoter regions [2,4,6,8,9,10]. Although many different KMT2A fusion partners have been characterized, some of the most common ones are AFF1/AF4, MLLT10/AF10, MLLT3/AF9, and MLLT1/ENL [3]. For example, the cell lines MOLM-13 and MOLM-14 have the fusion protein KMT2A-AF9 [11]. AF9 then recruits the methyltransferase, Disruptor of Telomeric Silencing 1-Like (DOT1L) [5].

DOT1L is a lysine methyltransferase that catalyzes mono-, di-, and tri-methylations of histone H3 at lysine 79 (H3K79me1, me2, or me3) using S-adenosylmethionine (SAM) as a cofactor in its active site [12,13,14]. Prior ubiquitination of H2B is observed to increase H3K79 methylation efficiency. Genome-wide studies using chromatin immunoprecipitation (ChIP) and ChIP sequencing (ChIP-Seq) indicate that H3K79me2 is associated with active gene transcription in humans [13,15,16]. Knockdown studies of DOT1L show a complete loss of H3K79 methylation, indicating that DOT1L is the only known H3K79 methyltransferase [13].

DOT1L has a *C*-terminal non-specific DNA-binding domain that aids in its normal interactions with nucleosomes [12]. In AML, DOT1L is aberrantly recruited to inappropriate genomic locations through its interaction with AF9, ENL, AF10, and AF17 [15]. AF9 and ENL, readers that bind acetylated H3K9, recruit DOT1L to open chromatin regions [16,17,18]. The fusion protein KMT2A-AF9 aberrantly recruits DOT1L to genes that are typically the target of KMT2A, such as the *HOX* genes. This chimeric protein loses its H3K4 methylation function while gaining H3K79 methylation, resulting in high expression of HOX genes like *HOXA9*. The aberrantly high *HOXA9* expression results in MLL, and knockdown of *HOXA9* expression results in reduced proliferation and apoptosis [19]. DOT1L was hypothesized to directly interact with the carboxy-terminal domain of the largest RNAP II subunit during transcriptional elongation [13] because of its colocalization with RNAP II [20]. However, recent ChIP-seq, PRO-seq, and 4sUDRB-Seq analyses showed that DOT1L is unlikely to have a role in elongation; instead, it may be involved in transcriptional initiation [20].

Histone modifications associated with the transcriptionally active state of genes include H3K4me1, H3K4me3, and H3K27ac, among many others. For a subset of genes, active marks such as H3K4me3, H3K27ac, and H3K79me2 are associated with a broad stretch of the gene body starting at the 5′ end of the gene [21,22]. This was initially observed with H3K4me3 and was named the broad H3K4me3 domain [22,23]. We and others have shown that genes with the broadest H3K4me3 domains identify cancer-related genes that are specific to the cancer-cell type [24]. For example, Belhocine et al. identified genes with the broad H3K4me3 domain in human thymic T-cell populations versus in T-Acute Lymphoblastic Leukemia (T-ALL) samples [25]. Expressed genes marked with the broad H3K4me3 domain in normal cells were cell-identity genes and tumor-suppressor genes, whereas in cancer, genes marked in this way were oncogenes involved in critical cancer processes.

To date, H3K4me3 is the most-studied broad H3K4me3 domain. Recently, we reported that H3K4ac is also arranged in a broad domain that is associated with critical genes and regulatory regions involved in breast cancer [24]. In this bioinformatic study, we identified genes with the broadest H3K79me2 domain that were unique to different types of leukemic cells. We show that the genes with the broadest H3K79me2 domain have known roles in leukemia and identify genes that should be further characterized as potential drivers of leukemia.

## 2. Results

### 2.1. Identification of Genes with a Broad H3K79me2 Domain in Leukemic Cells

To determine whether human leukemic cells have a broad H3K79me2 domain, we first analyzed publicly available H3K79me2 ChIP-Seq data using a representative acute myeloid leukemia cell line (MLL-type) MOLM-13. The MOLM-13 cell line was derived from a 20-year-old male with acute monocytic leukemia (AML-M5a) resulting from the chromosomal insertion (11;9)(q23; p22p23) [11]. This cell line expresses the KMT2A-AF9 fusion protein, which aberrantly recruits the DOT1L H3K79 methyltransferase to *HOX* genes, resulting in their overexpression. Appropriate GEO files (GSE113191, GSE149183) were identified, and the FASTQ files were entered in the Partek Flow program. The MACS2 peak caller was set in the broad mode. Figure 1 shows that most of the H3K79me2 presented as a narrow peak at the 5′ end of the gene and a lesser number of H3K79me2 peaks were quite broad, up to 471,096 bp long. The gene with the broadest H3K79me2 domain was *JMJD1C* (chr10:63,163,632-63,634,728) (Appendix A), a known target of the KMT2A-AF9 fusion protein and an oncogene essential to the self-renewal of AML cells [26]. The breadth of the top 5% of broad H3K79me2 domains in MOLM-13 cells ranged from 25,846 to 471,096 bp.

Next, we repeated these analyses with cell lines representing chronic myeloid leukemia (K562), adult acute lymphoblastic leukemia (Loucy), and childhood acute lymphoblastic leukemia (DND-41). K562 cells were derived from a 53-year-old female patient in blast crisis. It was the first immortalized CML cell line, with a hypotriploid karyotype [27,28,29]. Loucy cells were derived from the peripheral blood of a 38-year-old European female adult with T-cell ALL with a t(16:20) and 5q- chromosomal aberrations [30]. DND-41 was derived from the peripheral blood of a 13-year-old male with T-cell ALL and carrying a p53 mutation [31]. None of the cell lines that we used have known mutations to DOT1L. We used the following GEO files—K562, GSE29611; Loucy, GSE96166; DND-41, GSE29611—and loaded the FASTQ files into Partek Flow for analysis as described for MOLM-13 cells. The breadth of the top 5% broadest H3K79me2 peaks varied among the cell lines (DND-41, 27,560–340,185 bp; K562, 18,304–191,997 bp; Loucy, 3241–8467 bp), with the breath of the H3K79me2 region being the shortest in the Loucy cell line (Appendix A). Other available H3K79me2 ChIP-Seq data for the Loucy cell line (GEO file GSE76745) were analyzed, and the maximal length of the broad H3K79me2 domains was found to be 3400 to 6077 bp. Together, these results are consistent with the Loucy cell line having the shortest of the broad H3K79me2 domains.

The gene with the broadest H3K79me2 domain in each cell line was *SPG7* (chr16:89,301,571-89,493,568) in K562; *PTPRC* (chr1:198,683,545-198,692,012) in Loucy and Neuroligin 4 X-linked *(NLGN4X)* (chrX:5,892,371-6,232,556) in DND-41. SPG7 matrix AAA peptidase subunit paraplegin (*SPG7*) codes for a mitochondrial metalloprotease protein and is upregulated in colorectal cancer [32]. Protein tyrosine phosphatase receptor-type C (*PTPRC*) gene, a member of the protein tyrosine phosphatase (PTP) family, is abnormally expressed in leukemic cells [33]. Neuroligin 4 X-linked (*NLGN4X*) codes for a type-B carboxylesterase/lipase protein family member and is associated with poor relapse-free survival in triple-negative breast cancer [34]. Although these genes have known roles in cancer, only the *PTPRC* gene has a documented role in leukemic cells.

GO-term analysis (biological process) for genes with the top 5% broad H3K79me2 domains in the four cell lines was carried out using Enrichr (Table 1) [35,36]. Phosphorylation as a biological process was prominent for genes in MOLM-13, DND-41, and K562 cells, whereas the Loucy cell line had genes with biological processes involved in RNA splicing.

In comparison, we determined the GO terms for genes that had the shortest H3K79me2 regions (bottom 5%). Protein phosphorylation was among the most significant biological processes for the four cell lines. The most significant GO terms for the bottom 5% of the genes in the four cell lines were as follows: MOLM-13, protein phosphorylation; DND-41, peptidyl-serine phosphorylation; K562, epidermal growth factor receptor-signaling pathway; Loucy, regulation of cell–matrix adhesion.

Publicly available RNA sequencing (RNA-Seq) data for the four cell lines were analyzed with Partek Flow. Appropriate RNA-Seq files were identified (K562-GSE90239, GSE78556, GSE140322; Loucy-GSE155337; DND41-GSE173867, GSE116873; MOLM-13-GSE113191, GSE181003). The RNA-Seq was done in duplicate or triplicate in each study. The FASTQ files were uploaded into the Partek Flow program. For DOT1L expression, the enzyme was expressed at a greater level in MOLM-13 cells compared to the other three cell lines (based on TPM: MOLM-13, 27.0; DND-41, 3.8; K562, 2.5; Loucy, 5.2). Following normalization using TPM (Appendix A), the transcript levels in each RNA-Seq study were binned into five groups. The genes in the top 5% broad H3K79me2 domains for each cell line were then cross-referenced with the transcript level of that gene. Approximately 60% of the genes with the broad H3K79me2 domain had transcript levels in the top 40% (K562, 61%; Loucy, 61%; DND-41, 56%; MOLM-13, 61%), and about 80% of the genes with the broad H3K79me2 domain were in the top 60% (K562, 88%; Loucy, 81%; DND-41, 84%; MOLM-13, 89%). These observations show that the genes with the broadest H3K79me2 domains tend to exhibit higher transcript levels.

The top 5% broad H3K79me2 domains in the four leukemic cell lines that were unique to that cell line were identified (Appendix A) and are displayed in Figure 2. The GO terms for these genes (biological process) included “negative regulation of myeloid-cell differentiation (MOLM-13)”, “mitotic-chromosome condensation (DND-41)”, “regulation of organelle organization (K562)”, and “RNA splicing (Loucy)” (Table 2).

Figure 3 shows H3K79me2 tracks for genes that were in the top 5% of broadest H3K79me2 domains. A broad H3K79me2 region was observed over the *DACH1* gene in MOLM-13 cells but not in the other cell lines (Figure 3A). The dachshund-family transcription factor 1 (*DACH1*) gene is upregulated by the KMT2A-AF9 fusion protein [37]. DACH1 is a coactivator of HOXA9 and is involved in myeloid leukemogenesis. An intense H3K79me2 region was present over the zinc-finger protein 521 (*ZNF521*) gene in MOLM-13 and to a lesser extent in Loucy (Figure 3B). DND-41 and K562 cells did not have H3K79me2 over the *ZNF521* gene. All four lines showed a sharp H3K79me2 peak at the 5′ end of the *SS18* gene, which is next to the *ZNF521* gene. In contrast, the *PSMA8* gene did not have an associated H3K79me2 peak. KMT2A-AF9 directly interacts with the *ZNF521* promoter, activating its transcription [38]. ZNF521 is a critical effector of KMT2A fusion proteins in blocking myeloid differentiation and can be used as a potential therapeutic target for this leukemia subtype [38]. DND-41 and, to a much lesser extent in Loucy, has a broad H3K79me2 domain over the *NLGN4X* gene (Figure 3C). Neuroligin 4 X-linked (*NLGN4X*) is a protein-coding gene that encodes a member of the type-B carboxylesterase/lipase protein family [39]. NLGN3/4X activates actin regulators p21-activated kinase 1 and cofilin, which are involved in mediating the effects of adhesion proteins on actin filaments, growth cones, and neuritogenesis [40]. The *HIVEP3* gene has a broad H3K79me2 region in DND-41 cells (Figure 3D). *HIVEP3* encodes a member of the human immunodeficiency virus type 1 enhancer-binding protein family [39]. It contains multiple zinc-finger and acid-rich (ZAS) domains, as well as serine–threonine-rich regions [39]. The ZAS protein family has been implicated in regulating gene expression of HIV-1 long-terminal repeat and other genes [41]. *HIVEP3* is a key regulator that controls (both transcriptionally and post-transcriptionally) innate-like T-lymphocyte-cell development [42]. Next to the *HIVEP3* gene is the *FOXJ3* gene, which has an H3K79me2 peak at the 5′ end of the gene in the four cell lines. Loucy has a broad H3K79me2 region over the Sprouty RTK-signaling antagonist 1 (*SPRY1*) gene which was not observed in the other cell lines (Figure 3E). *SPRY1* is involved in the negative regulation of the fibroblast growth-factor receptor-signaling pathway [39]. It is one of the many genes overexpressed in p185^BCR-ABL^-positive ALL [43]. *SPRY1* has been observed to be markedly overexpressed in cells of AML patients [44]. AML patients with high *SPRY1* expression had poor prognoses [44]. The *SGMS1* gene in K562 cells had two broad H3K79me2 regions that were not observed in the other cell lines (Figure 3F). Sphingomyelin synthase 1 is encoded by the *SGMS1* gene and metabolizes ceramides into sphingomyelin, the most abundant sphingolipid in mammalian cells [45]. Together these observations identified cell type-specific genes that have broad H3K79me2 domains. Several of these genes have known roles in leukemia.

### 2.2. MOLM-13 and HOXA9 Cluster

In MOLM-13 cells, the *HOXA9* gene is immersed in a broad H3K79me2 domain (Figure 4). The H3K79me2 tracks and RNA tracks for the *HOXA9*, *HOXA10*, and *HOXA11* genes in MOLM-13 cells are shown in Appendix A. This gene region was identified as HOXA11-AS1_3 by Partek Flow and as being unique to MOLM-13 with a broad H3K79me2 domain in the top 5%. This domain was also rich in DOT1L. DOT1L appears to be distributed over two sections in this domain. Of interest, the *HOXA9* broad H3K79me2 domain has several genes with known roles in leukemia (Appendix A). *HOXA*-cluster members, consisting of 11 genes and non-coding RNA elements, are widely expressed in myeloid cells [46,47]. Among the *HOXA* gene family are genes encoding DNA-binding proteins that widely control embryo segmentation and later developmental events. Several studies reported that overexpression of *HOXA* genes is directly involved in carcinoma progression, especially in acute myeloid leukemia and lymphoid leukemia. In this cluster, *HOXA9*, *10*, and *11* had the highest transcript levels in MOLM-13 cells (Appendix A). In addition to the transcription-factor-coding genes, several genes express long non-coding RNAs, many of which are oncogenic lncRNAs. Both *HOTTIP* and *HOTAIRM1*, for example, have known roles in leukemia [48,49].

Another interesting feature of the *HOXA9* broad H3K79me2 domain is that the domain is hypersensitive to DNase I digestion, which identifies accessible chromatin regions (Figure 4). The DNase I hypersensitivity covers the entire *HOXA9* broad H3K79me2 domain and regions on either side of the domain. This observation suggests that this region has an unstable chromatin structure.

The *HOXA-AS3* gene region was also identified as having a unique broad H3K79me2 domain in Loucy cells (Figure 5). This region has a very similar broad H3K79me2 domain over the *HOXA9* gene cluster to what was observed for MOLM-13 cells; however, MACS2 called these peaks differently (*HOXA11-AS1_3* for MOLM-13 and *HOXA-AS* for Loucy). The SET-NUP214 fusion protein is expressed in the Loucy cell line [50]. This fusion protein binds to the promoters of the *HOXA9* and *HOXA10* genes and recruits DOT1L to these sites [50,51]. Although the mechanism by which DOT1L is recruited to the *HOXA9* cluster is different in MOLM-13 and Loucy cells, the result is similar, with the domain acquiring a broad H3K79me2 domain.

### 2.3. Broad H3K79me2 Domain in AML Cell Lines

We next explored whether different AML cells (MLL-type) that had the KMT2A fusion protein would have different sets of genes with the broadest H3K79me2 domain. FASTQ files were obtained from available public datasets (MV4-11, GSE79899; THP-1, GSE79899, MOLM-13, GSE149183). The breadth of the top 5% of broadest H3K79me2 peaks varied among the cell lines (MV4-11, 3410–11,073 bp; THP-1, 2601–6194 bp; MOLM-13, 25,846–471,096 bp) (Appendix A). The *JMJD1C* gene and *HOXA* (including *HOXA9*) gene clusters were associated with a broad H3K79me2 domain in these three AML (MLL-type) cell lines. The HOXA gene cluster was considerably shorter in THP-1 (45 kb) and MV4-11 (30 kb) cells than in MOLM-13 (116 kb) and Loucy (111 kb) cells. The *DACH1* gene had a broad H3K79me2 domain in MOLM-13 and THP-1, but not in MV4-11 cells. As with MOLM-13 cells, *NLGN4X, HIVEP3, SPRY1*, and *SGMS1* did not have an H3K79me2 signature in THP-1 or MV4-11.

GO-term analysis (biological process) of genes with the top 5% of broad H3K79me2 domains in the three AML cell lines was conducted using Enrichr (Table 1 and Table 3). Although the GO term “phosphorylation” was prominent for MOLM-13 genes, MV4-11 and THP-1 had genes involved in RNA splicing.

The top 5% of broad H3K79me2 domains in the three AML cell lines that were unique to that cell line were identified (Figure 6, Appendix A). These analyses found the following genes in the four cell lines: MV4-11, 1099; THP-1, 363, MOLM-13, 886. The GO terms for these genes (biological process) included “protein phosphorylation (MOLM-13)”, “nuclear import (MV4-11),” and “mitotic spindle organization (THP-1)” (Table 4).

Genes with the broad H3K79me2 domain and listed as present in the top 5% of an AML cell line include CUB And Sushi Multiple Domains 3 (*CSMD3*, MV4-11), Twist Family BHLH Transcription Factor 1 (*TWIST1*, THP-1), and Ephrin A5 (*EFNA5*, MOLM-13). The *CSMD3* gene has been investigated for its possible role in cancer progression [52]. The *TWIST1* gene is involved in epithelial–mesenchymal transition and is involved in several cancers, including AML [53,54]. The *EFNA5* gene plays a role in several cancers, including ovarian cancer and prostate cancer [55,56]. It will be of interest to further explore the role of these genes in AML. In summary, the three AML (MLL-type) cell lines have genes in common and genes that are distinct with the broad H3K79me2 domain.

## 3. Discussion

DOT1L-mediated methylation of H3K79 is an active mark that is involved in transcription initiation [20]. Mass spectrometry analyses of cellular H3K79 unmodified and methyl forms in a variety of mammalian cells (THP-1, U937, HL60, mouse thymus) showed that K79 is about 88–95% unmodified, 4.4–10.1% is K79me1, and 0.5–1.6% is in K79me2, whereas H3K79me3 is undetectable [57,58]. The extent of H3K79 methylation increases with acetylation of H3, with K79me1 predominating in the tetra-acetylated state [59]. DOT1L methylation of H3K79 is enhanced when the nucleosome has ubiquitinated H2B, which is dependent upon ongoing transcription [60,61]. Acetylation of H3 is also linked to ongoing transcription [62]. Thus, H3K79 methylation is augmented with H3 acetylation and H2B ubiquitination (K120), modifications that are both dependent upon ongoing transcription.

Several studies concluded that H3K79me2 is an indicator of genes involved in leukemogenesis [63,64]. Inhibition of DOT1L activity reduces *HOXA9* expression and leukemic cell survival [14]. Together, these observations show the importance of H3K79me2 catalyzed by DOT1L. However, DOT1L is also involved in the destabilization of the nucleosome, which is independent of its catalytic activity [65]. The enhanced chromatin accessibility over the 100 kb *HOXA9* gene domain, which is evinced by the DNase sensitivity data in Figure 4, is likely the result of multiple factors that destabilize nucleosome structure, including DOT1L itself, histone modifications such as ubiquitinated H2B and acetylation, and chromatin remodelers. We envisage a dynamic process in which the dissolution of nucleosomes with heavily modified histone octamers would transiently present the accessible DNA to a host of proteins (such as transcription factors), followed by the reassembly of the nucleosome with unmodified histones and specific histone variants (such as the replacement histone variant H3.3) [21,66].

With regards to RNA splicing, it has been reported that DOT1L-mediated H3K79me2 is involved in alternate splicing (skipping exon and alternate 3′ splice site) [67]. Exon skipping was prevalent in AML (MLL) cell types. For genes with a broad H3K79me2 domain that participate in alternative splicing, this histone modification could have a significant impact on the transcript produced and contribute to leukemic transformation [67].

Several histone modifications (H3K4me3, H3K4ac, H3K27ac, H3K79me2) associated with transcribed genes exhibit a broad distribution [23,24,26]. In normal cells, the broadest of the H3K4me3 domains are associated with genes involved in cell identity, whereas in cancer cells, the genes marked with the broadest H3K4me3 domain are cancer-related [21,25]. We recently showed that the genes marked with the broadest H3K4ac domain identify genes important in different breast-cancer types [24]. Here, we applied a similar strategy to identify genes important to different leukemic cell types by finding which genes had the broadest H3K79me2 domain that was present solely in one of the four cell types.

In our initial analyses of genes with H3K79me2 ChIP-Seq data for MOLM-13 cells, the *JMJD1C* gene was identified as having the broadest H3K79me2 domain. JMJD1C is an H3K9 demethylase with a known oncogenic role in AML and, in particular, those caused by insertions and translocations of KMT2A. *JMJD1C* is overexpressed in cells with the KMT2A-AF9 and other fusions and is required for self-renewal of leukemic cells [68]. However, *JMJD1C* has a role in other AML cells, where it is required by RUNX1–RUNX1T1 to increase cell proliferation [69].

The breadths of the H3K79me2 domains in Loucy and THP-1 cells were considerably shorter than those of the other leukemia cells studied. The reason(s) for the shorter H3K79me2 domain, whether biological or technical, requires further investigation. 

Identifying genes that have the broadest H3K79me2 domain in only one of the four leukemic cell lines (MOLM-13, K562, Loucy, DND-41) produced an informative list of genes. In MOLM-13 cells, for example, *DACH1* and *ZNF21* genes are shown to have broad H3K79me2 domains. Both genes play critical roles in AML and are potential therapeutic targets. Several genes in MOLM-13 cells identified in this analysis have been documented as playing critical roles in AML with the KMT2A-AF9 translocation—for example, *RUNX2*, *MEIS1*, *HOXA9*, and *PBX3* [70]. Except for *PBX3*, these genes are unique in having the broadest H3K79me2 domain in MOLM-13 cells. *PBX3* is also in the top 5% broadest H3K79me2 domains in Loucy cells, but not in DND-41 or K562 cells.

There are several limitations to this study. Firstly, a limitation to the analyses of broad H3K79me2 domains and broad modified-histone domains in general is the quality of the antibody used [71]. In reviewing the GEO files used in these studies, the source of the anti-H3K79me2 antibody was not always listed. For other GEO entries, there is the concern that the anti-H3K79me2 antibody cross-reacts with H3K79me1, which is more abundant than H3K79me2 [71]. In future studies of the broad H3K79me2 domain, it is critical that the antibody against H3K79me2 undergo a rigorous validation [71]. Secondly, there are several limitations to our observation of the top 5% of broad H3K79me2 domains tending to have a higher level of gene expression. The levels of transcripts are steady-state, and the cellular level of a particular transcript would depend on the gene’s rate of transcription, pre-mRNA splicing, transcript turnover, and many other factors. To accurately and directly determine the impact of the chromatin structure of the broad H3K79me2 on a particular gene, transcription needs to be measured by a nuclear run-on assay. It was noted that one of the features of genes with the broad H3K4me3 domain is transcriptional consistency [22]. Whether genes with the broad H3K79me2 domain exhibit transcriptional consistency remains to be determined. Thirdly, future studies need to be done with leukemia-patient samples.

One of the most interesting broadest H3K79me2 domains includes the *HOXA9* gene, which is found in MOLM-13 and Loucy cell lines, which can be explained by the recruitment of DOT1L to *HOXA9* genes but through different mechanisms. The *HOXA9* genes in K562 and DND-41 cells do not have an associated broad H3K79me2 domain. In MOLM-13 cells, DOT1L is recruited to HOXA9 via the oncogenic fusion protein KMT2A-AF9, whereas in Loucy cells, SET-NUP214 recruits DOT1L. The *HOXA9* domain, which is rich in CpG islands, can bind to the CXXC domain of KMT2A-AF9, resulting in the recruitment of DOT1L to these regions, and such binding prevents methylation of these CpG sequences [72]. The broad H3K79me2 domain contains several transcription-factor-coding genes (e.g., HOXA10) and oncogenic lncRNAs. In future studies, targeting multiple sites in this 100 kb H3K79me2 domain through epigenetic-editing methods may result in the silencing of several oncogenic players involved in leukemia. 

## 4. Conclusions

Genes associated with the broadest active histone mark (e.g., H3K4me3, H3K4ac, H3K79me2) have essential functions in cells, and this epigenetic signature can identify genes important in normal and disease states. Through approaches that we describe here, gene panels can be generated to aid in the diagnosis and treatment of disease.

## 5. Materials and Methods

### 5.1. Data Collection 

Analyses were conducted using publicly available ChIP-Seq data from the Gene Expression Omnibus (GEO) database. The GEO files for the H3K79me2 ChIP-Seq files for the following cell lines were as follows: MOLM-13 (GSE149183, GSE43063, GSE113191), MOLM-14 (GSE76745), K562 (GSE29611), Loucy (GSE96166), DND-41 (GSE29611), MV4-11 (GSE79899), and THP-1 (GSE79899). The GEO files for the DNase I-Seq files for MOLM-13 were from GSE149183. DOT1L ChIP-Seq files for MOLM-13 were from GSE149183. RNA-Seq files for the leukemic cell lines were from the following sources: K562, GSE90239, GSE78556, GSE140322; Loucy, GSE155337; DND-41, GSE173867, GSE116873; MOLM-13, GSE113191, GSE181003.

### 5.2. ChIP Sequence Analyses (MACS2)

FASTQ files were obtained from the European Nucleotide Archive and downloaded to the Partek Flow analytical program (Partek Incorporated, 100 Chesterfield Business Parkway, Chesterfield, Missouri 63005). The Bowtie 2 aligner was used, with sequences being aligned against the human hg38 assembly. Identification of broad H3K79me2 peaks was carried out using the MACS2 broad-peak function. The peaks were annotated using Ensembl transcript release 91.

### 5.3. Broad H3K79me2 Domain

The top 5% of the broadest domains were extracted from the H3K79me2 peaks identified by the MACS2 peak caller. We used online tool VENNY 2.1 to classify genes per cell type and to identify the genes that were unique to each cell line. The VENNY 2.1 tool was then applied to identify the unique genes with the broadest H3K79me2 domains. We extracted these exclusive genes and performed gene ontology (GO) (biological process) with Enrichr [35,36].

### 5.4. RNA Sequence Analyses

FASTQ files for the cell lines were obtained from the European Nucleotide Archive and downloaded to the Partek Flow analytical program (Partek Incorporated, 100 Chesterfield Business Parkway, Chesterfield, Missouri 63005). The HISAT 2.1.0 aligner was used, with sequences being aligned against the human hg38 assembly. Quantification was completed using the Partek Flow “quantify to annotation model (Partek E/M)” and the Ensembl transcript release 104. Normalization was carried out using the TPM method.

## Figures and Tables

**Figure 1 cells-11-02830-f001:**
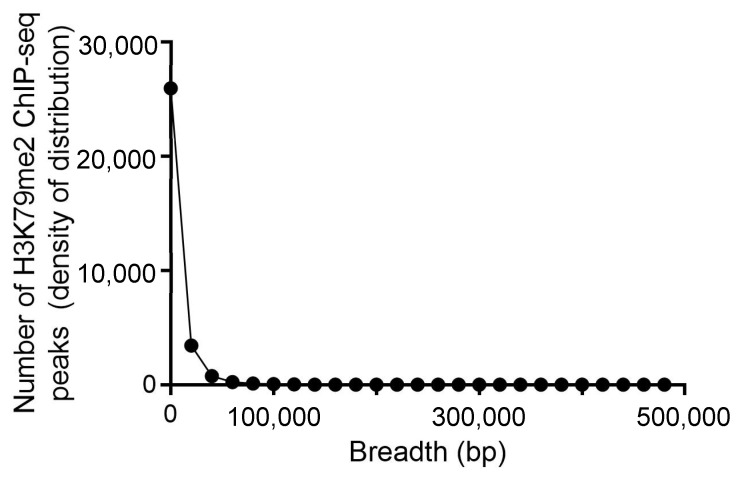
Density distribution of H3K79me2 peaks in MOLM-13 cells. The panel shows the density distribution of H3K79me2 peaks identified by the MACS2 peak caller.

**Figure 2 cells-11-02830-f002:**
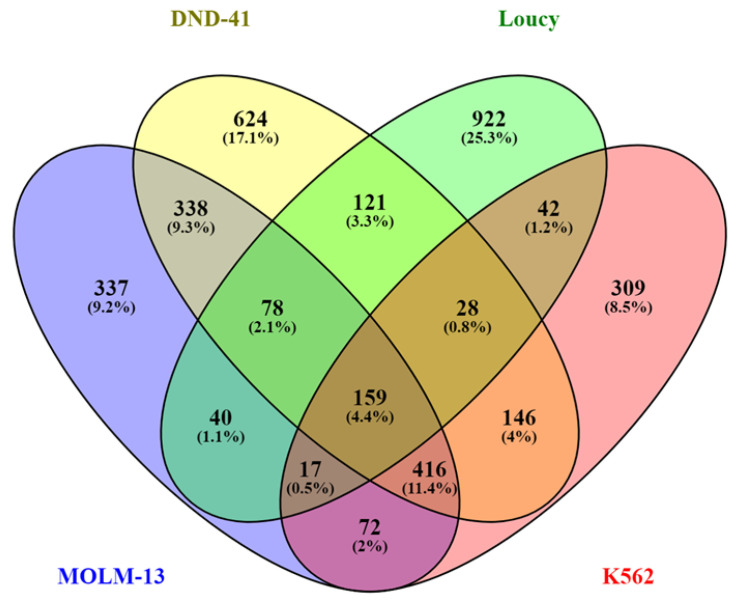
Distribution of genes with the broad H3K79me2 domain in MOLM-13, K562, Loucy, and DND-41 cells. The number of genes that are unique and in common in the top 5% of genes with the broad H3K79me2 among the four cell lines is shown.

**Figure 3 cells-11-02830-f003:**
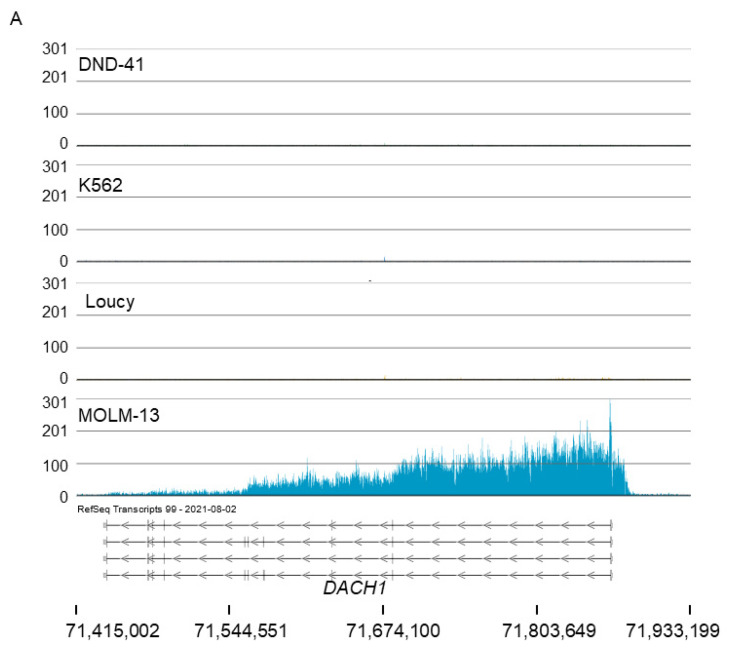
H3K79me2 ChIP-Seq tracks of genes with the broad H3K79me2 domain. The panel shows the H3K79me2 ChIP-Seq tracks for the four cell lines. (**A**) *DACH1*, (**B**) *ZNF21*, (**C**) *NLGN4X*, (**D**) *HIVEP3*, (**E**) *SSPRY1*, (**F**) *SGMS1*.

**Figure 4 cells-11-02830-f004:**
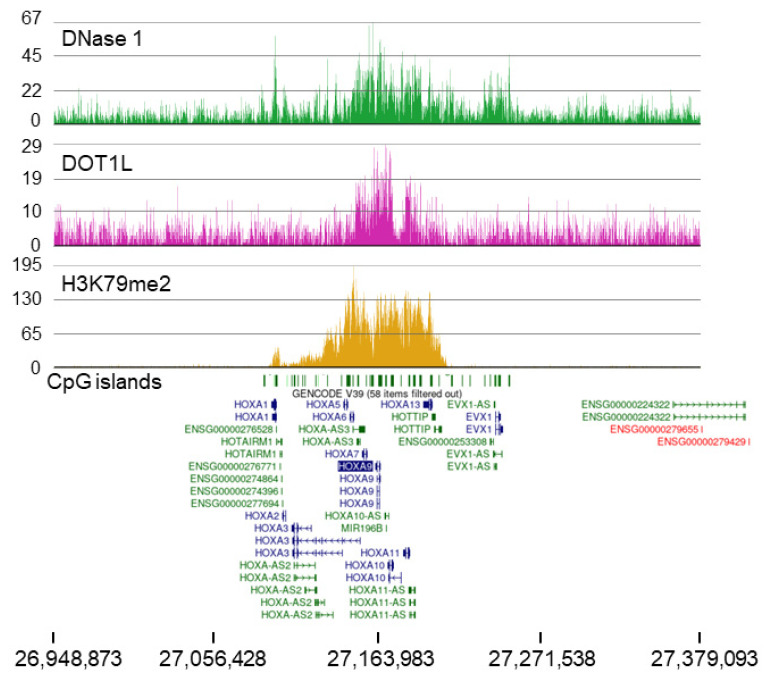
Broad H3K79me2 domain with the *HOXA9* gene in MOLM-13 cells. ChIP-Seq tracks for DOT1L and H3K79me2 are shown. The DNase 1-Seq track shows the chromatin accessibility of the broad H3K79me2 domain with the *HOXA9* gene in MOLM-13 cells. Through the CXXC domain, KMT2A-AF9 could be recruited to the numerous CpG islands present in this genomic region. KMT2A-AF9 recruits DOT1L, which methylates H3K79.

**Figure 5 cells-11-02830-f005:**
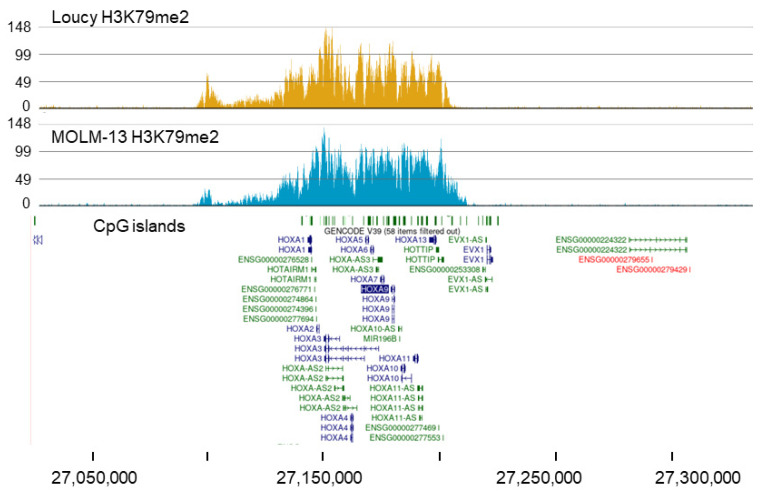
Broad H3K79me2 domain with the *HOXA9* gene in Loucy cells. The panel shows the H3K79me2 ChIP-Seq tracks for *HOXA9* in MOLM-13 and Loucy cells.

**Figure 6 cells-11-02830-f006:**
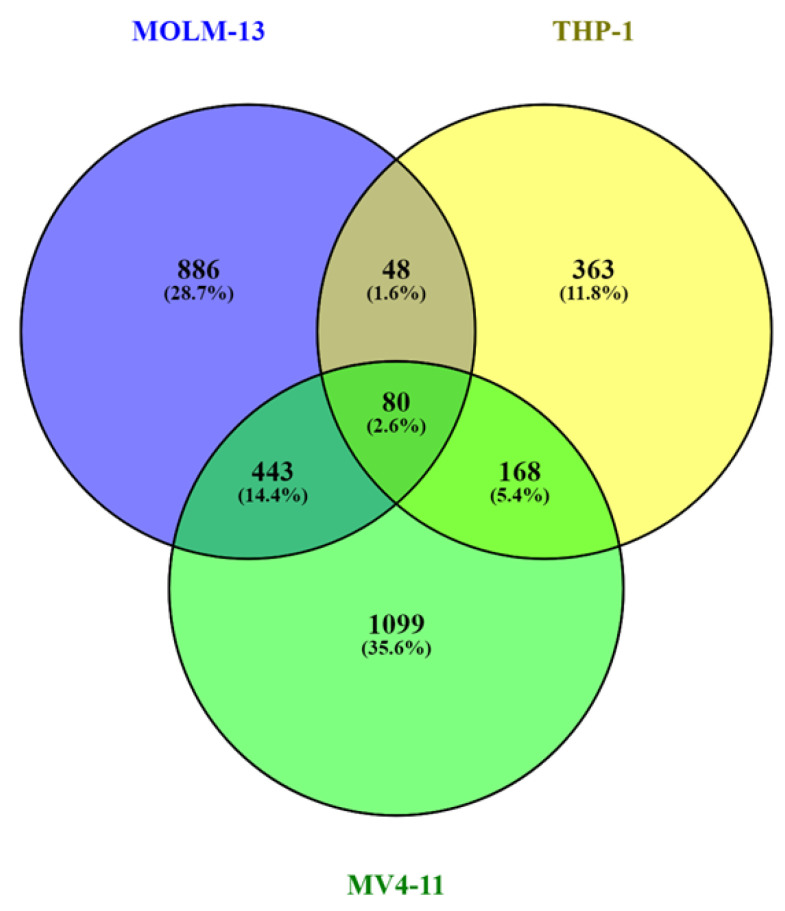
Distribution of genes with the broad H3K79me2 domain in MVP-11, THP-1, and MOLM-13. Genes unique and in common in the top 5% of genes with the broad H3K79me2 among the three AML cell lines.

**Table 1 cells-11-02830-t001:** GO terms (biological processes) for genes with the broadest H3K79me2 domains in leukemic cells.

MOLM-13 Top 5%					
Term	Overlap	*p*-Value	Adjusted *p*-Value	Odds Ratio	Combined Score
Phosphorylation (GO:0016310)	70/400	5.35 × 10^−12^	2.09 × 10^−8^	2.785	72.295
Positive regulation of transcription, DNA-templated (GO:0045893)	144/1183	4.52 × 10^−10^	8.81 × 10^−7^	1.848	39.758
Protein phosphorylation (GO:0006468)	75/496	1.13 × 10^−9^	1.47 × 10^−6^	2.336	48.130
Protein dephosphorylation (GO:0006470)	32/139	4.05 × 10^−9^	3.95 × 10^−6^	3.869	74.773
**DND-41 Top 5%**					
**Term**	**Overlap**	***p*-value**	**Adjusted *p*-value**	**Odds Ratio**	**Combined Score**
Mitotic cell-cycle phase transition (GO:0044772)	59/209	1.06 × 10^−14^	4.49 × 10^−11^	3.812	122.675
G1/S transition of mitotic cell cycle (GO:0000082)	31/85	1.65 × 10^−11^	3.50 × 10^−8^	5.510	136.810
Ubiquitin-dependent protein catabolic process (GO:0006511)	71/354	1.10 × 10^−9^	1.30 × 10^−6^	2.429	49.941
Protein dephosphorylation (GO:0006470)	38/139	1.54 × 10^−9^	1.30 × 10^−6^	3.615	73.357
**K562 Top 5%**					
**Term**	**Overlap**	***p*-value**	**Adjusted *p*-value**	**Odds Ratio**	**Combined Score**
Negative regulation of translation (GO:0017148)	26/90	7.53 × 10^−12^	2.78 × 10^−8^	6.549	167.727
Regulation of translation (GO:0006417)	36/178	7.56 × 10^−11^	1.39 × 10^−7^	4.105	95.670
Phosphorylation (GO:0016310)	57/400	7.38 × 10^−10^	9.08 × 10^−7^	2.711	57.009
Negative regulation of gene expression (GO:0010629)	48/322	3.72 × 10^−9^	3.43 × 10^−6^	2.846	55.243
**Loucy Top 5%**					
**Term**	**Overlap**	***p*-value**	**Adjusted *p*-value**	**Odds Ratio**	**Combined Score**
mRNA splicing, via spliceosome (GO:0000398)	55/274	1.11 × 10^−12^	2.52 × 10^−9^	3.413	93.961
RNA splicing via transesterification reactions with bulged adenosine as nucleophile (GO:0000377)	52/251	1.28 × 10^−12^	2.52 × 10^−9^	3.547	97.139
mRNA processing (GO:0006397)	57/300	4.58 × 10^−12^	6.01 × 10^−9^	3.188	83.248
RNA metabolic process (GO:0016070)	34/133	2.55 × 10^−11^	2.51 × 10^−8^	4.626	112.835

**Table 2 cells-11-02830-t002:** GO terms (biological processes) for genes with the broadest H3K79me2 domains unique to one of the four cell lines.

MOLM-13					
Term	Overlap	*p*-Value	Adjusted *p*-Value	Odds Ratio	Combined Score
Negative regulation of myeloid-cell differentiation (GO:0045638)	4/22	4.56 × 10^−4^	0.403777	13.10978	100.8647
Phosphorylation (GO:0016310)	17/400	4.81 × 10^−4^	0.403777	2.674282	20.42911
Regulation of cellular senescence (GO:2000772)	4/31	0.001739	0.573737	8.735847	55.50902
Negative regulation of NF-kappaB transcription-factor activity (GO:0032088)	6/79	0.002167	0.573737	4.864462	29.83979
**DND-41**					
**Term**	**Overlap**	***p*-value**	**Adjusted *p*-value**	**Odds Ratio**	**Combined Score**
Mitotic-chromosome condensation (GO:0007076)	6/27	1.52 × 10^−4^	0.134681	8.94822	78.65524
DNA dealkylation (GO:0035510)	4/10	1.70 × 10^−4^	0.134681	20.82796	180.8268
DNA dealkylation involved in DNA repair (GO:0006307)	4/10	1.70 × 10^−4^	0.134681	20.82796	180.8268
Mitotic-cell-cycle phase transition (GO:0044772)	17/209	3.14 × 10^−4^	0.18717	2.798325	22.56889
**K562**					
**Term**	**Overlap**	***p*-value**	**Adjusted *p*-value**	**Odds Ratio**	**Combined Score**
Regulation of organelle organization (GO:0033043)	7/80	2.38 × 10^−4^	0.207771	6.229066	51.9768
Regulation of cytoskeleton organization (GO:0051493)	8/112	3.51 × 10^−4^	0.207771	5.005622	39.81809
Negative regulation of stress-fiber assembly (GO:0051497)	4/24	4.65 × 10^−4^	0.207771	12.89902	98.97629
Establishment of endothelial intestinal barrier (GO:0090557)	3/11	5.50 × 10^−4^	0.207771	24.12132	181.0597
**Loucy**					
**Term**	**Overlap**	***p*-value**	**Adjusted *p*-value**	**Odds Ratio**	**Combined Score**
RNA splicing, via transesterification reactions with bulged adenosine as nucleophile (GO:0000377)	31/251	6.03 × 10^−7^	0.001095	2.982339	42.71137
Mitotic-spindle organization (GO:0007052)	23/157	8.91 × 10^−7^	0.001095	3.616888	50.38532
mRNA splicing via spliceosome (GO:0000398)	32/274	1.39 × 10^−6^	0.001095	2.798551	37.75151
Gene expression (GO:0010467)	38/356	1.40 × 10^−6^	0.001095	2.535929	34.17959

**Table 3 cells-11-02830-t003:** GO terms (biological processes) for genes with the broadest H3K79me2 domains in AML cell lines.

MV4-11 Top 5%					
Term	Overlap	*p*-Value	Adjusted *p*-Value	Odds Ratio	Combined Score
mRNA splicing via spliceosome (GO:0000398)	64/274	5.72 × 10^−13^	2.37 × 10^−9^	3.178282	89.59665
mRNA processing (GO:0006397)	66/300	4.57 × 10^−12^	9.48 × 10^−9^	2.940925	76.79013
RNA processing (GO:0006396)	46/179	3.34 × 10^−11^	4.61 × 10^−8^	3.584974	86.48194
RNA splicing via transesterification reactions with bulged adenosine as nucleophile (GO:0000377)	56/251	1.07 × 10^−10^	1.11 × 10^−7^	2.983586	68.49088
**THP-1 top 5%**					
**Term**	**Overlap**	***p*-value**	**Adjusted *p*-value**	**Odds Ratio**	**Combined Score**
mRNA splicing via spliceosome (GO:0000398)	34/274	3.18 × 10^−11^	8.11 × 10^−8^	4.32956	104.6518
RNA splicing via transesterification reactions with bulged adenosine as nucleophile (GO:0000377)	32/251	5.98 × 10^−11^	8.11 × 10^−8^	4.456271	104.9003
mRNA processing (GO:0006397)	33/300	1.47 × 10^−9^	1.33 × 10^−6^	3.765912	76.5817
RNA metabolic process (GO:0016070)	19/133	7.67 × 10^−8^	5.20 × 10^−5^	5.007031	82.0337

**Table 4 cells-11-02830-t004:** GO terms (biological processes) for genes with broadest H3K79me2 domains unique to an AML cell line.

MV4-11					
Term	Overlap	*p*-Value	Adjusted *p*-Value	Odds Ratio	Combined Score
Import into nucleus (GO:0051170)	14/77	6.81 × 10^−5^	0.147625	3.858269	36.98274
Nucleotide-excision repair (GO:0006289)	16/105	1.92 × 10^−4^	0.147625	3.122745	26.72998
Positive regulation of establishment of protein localization to mitochondrion (GO:1903749)	11/56	1.99 × 10^−4^	0.147625	4.236438	36.0943
mRNA splicing via spliceosome (GO:0000398)	30/274	2.59 × 10^−4^	0.147625	2.145831	17.72484
**THP-1**					
**Term**	**Overlap**	***p*-value**	**Adjusted *p*-value**	**Odds Ratio**	**Combined Score**
Microtubule cytoskeleton organization involved in mitosis (GO:1902850)	10/128	1.17 × 10^−4^	0.097514	4.685985	42.43785
Mitotic-spindle organization (GO:0007052)	11/157	1.43 × 10^−4^	0.097514	4.171875	36.93698
Positive regulation of double-strand break repair via nonhomologous end joining (GO:2001034)	4/16	1.63 × 10^−4^	0.097514	18.22191	158.8791
Nuclear-membrane reassembly (GO:0031468)	6/54	4.25 × 10^−4^	0.164331	6.858894	53.24899
**MOLM-13**					
**Term**	**Overlap**	***p*-value**	**Adjusted *p*-value**	**Odds Ratio**	**Combined Score**
Phosphorylation (GO:0016310)	45/400	9.90 × 10^−9^	2.98 × 10^−5^	2.827469	52.11112
Protein phosphorylation (GO:0006468)	44/496	1.02 × 10^−5^	0.015305	2.157547	24.80196
Regulation of intracellular-signal transduction (GO:1902531)	38/437	6.19 × 10^−5^	0.062065	2.101864	20.36704
Transmembrane-receptor protein tyrosine kinase-signaling pathway (GO:0007169)	35/404	1.28 × 10^−4^	0.096379	2.089284	18.72459

## Data Availability

The study was done with publicly available datasets which can be obtained through Gene Expression Omnibus.

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
