# Peer review of "Bioinformatic Analyses of Broad H3K79me2 Domains in Different Leukemia Cell Line Data Sets"

_cells, 2022, doi:10.3390/cells11182830_

Round 1

Reviewer 1 Report

In this manuscript, Sharma et al identify genes enriched with broad H3K79me2 domains as involved in leukemia, along with decreased chromatin accessibility. They predict that these two features would mark genes that could be of use as biomarkers in diagnosis of leukemia. Overall, this is a very interesting finding that will be of broad interest to readers. I offer the following comments to assist the authors in improving their manuscript.

Comments: 

1. This is overall a well-written manuscript, describing analyses from existing datasets.

2. Figure 1. It would be of interest to know which categories of genes lie under the narrow peaks of H3K9Me2 as well. Are these GO terms different from the broad peaks? In other words, is there a statistically significant enrichment of the GO terms in the broad peaks relative to a random group of peaks and/or narrow peaks?

3. Based on the RNA-seq data, is there a difference in DOT1L expression in these 4 cell lines?

4. Is DOT1L mutated in these cell lines? DOT1L is frequently involved in chromosomal translocations itself, in MLL. This should be investigated and/or discussed.

5. Another important discussion point is that DOT1L and H3K79Me2 are correlated during alternative splicing. (for instance, see PMID: 29665865). In fact, splicing shows up as one of the GO categories in the current study. This is something that should be discussed.

Overall, I commend the authors for a very interesting and informative study.

Author Response

The revised manuscript is attached.

1. This is overall a well-written manuscript, describing analyses from existing datasets.

Response: The authors thank the reviewer for this comment.

  1. Figure 1. It would be of interest to know which categories of genes lie under the narrow peaks of H3K9Me2 as well. Are these GO terms different from the broad peaks? In other words, is there a statistically significant enrichment of the GO terms in the broad peaks relative to a random group of peaks and/or narrow peaks?

Response: We added a paragraph on lines 144 – 149 that lists the GO terms of the genes with the shortest H3K79me2 domains. To address the comments of reviewer 4, we revised the tables showing the GO terms and in doing so only used Enrichr.

  1. Based on the RNA-seq data, is there a difference in DOT1L expression in these 4 cell lines?

Response: We added an answer to this question on line 155. DOT1L is expressed in all cell lines but has the highest expression in MOLM-13 cells.

  1. Is DOT1L mutated in these cell lines? DOT1L is frequently involved in chromosomal translocations itself, in MLL. This should be investigated and/or discussed.

Response: To address this comment, we added a sentence on line 120. “None of the cell lines that we used have known mutations to DOT1L.”

  1. Another important discussion point is that DOT1L and H3K79Me2 are correlated during alternative splicing. (for instance, see PMID: 29665865). In fact, splicing shows up as one of the GO categories in the current study. This is something that should be discussed.

Response: Thanks for this comment. We added this important observation in the revised Discussion in the paragraph starting on line 353.

Reviewer 2 Report

In this study, the authors have identified a panel of genes with a broad H3K79me2 domain have known roles in Leukemia cells and thus could be used in the diagnosis and treatment of this disease. This is a very interesting study showing preliminary results that have to be addressed with additional data and discussion.

-          The study only shows the analysis of the publicly available data, but do not provide any functional evidence to show the relevance of broad H3K79me2 in leukemia development and progression.

-          The RNA-seq analysis indicate that the genes with broad H3K79me2 tend to exhibit higher transcript levels. However, the correlation between RNA expression for the broadest H3K79me2 domains are not shown.

-          In order to confirm the relevance of the study in the leukemia patients, the validation of the analysis have to be done both in the cell lines and sample patients.

Author Response

The revised manuscript is attached.

1. The study only shows the analysis of the publicly available data, but do not provide any functional evidence to show the relevance of broad H3K79me2 in leukemia development and progression.

Response: Throughout the article, we highlight the genes with the broadest H3K79me2 that have roles in leukemia (for example, the section 2.2 on MOLM-13 and the HOXA9 cluster (line 222).

  1. The RNA-seq analysis indicate that the genes with broad H3K79me2 tend to exhibit higher transcript levels. However, the correlation between RNA expression for the broadest H3K79me2 domains are not shown.

Response: There are several limitations to the interpretation of RNA Seq data that are based on steady state levels of transcripts. We speak to this comment in the paragraph starting on line 343.

“Our analyses of the relationship between the top 5% broad H3K79me2 domains indicated that the genes with this epigenetic signature tended to have a higher level of ex-pression. There are several limitations to these analyses. The levels of transcripts are steady-state, and the cellular level of a particular transcript would depend on the gene’s rate of transcription, pre-mRNA splicing, transcript turnover, and many other factors. To directly determine accurately the impact of the chromatin structure of the broad H3K79me2 on a particular gene, transcription needs to be measured by a nuclear run-on assay. It is noted that one of the features of genes with the broad H3K4me3 domain was transcriptional consistency [23]. Whether genes with the broad H3K79me2 domain exhibit transcriptional consistency remains to be determined.”

  1. In order to confirm the relevance of the study in the leukemia patients, the validation of the analysis have to be done both in the cell lines and sample patients.

Response: We added analyses of two MLL-type AML cell lines (MV4-11 and THP-1) (the new section 2.3 “Broad H3K79me2 domain in AML cell lines”; line 263). Analyses of the GO terms (biological processes) for the genes with the broadest 3K79me2 domains are presented in Table 3 and 4.

Reviewer 3 Report

The manuscript submitted by Sharma et al., employed ChIP-Seq data to identify genes with a broad H3K79me2 domain in human leukemic cell lines. By reducing the genes with the broadest H3K79me2 domain the authors identify some cancer-related genes associated with each leukemic cell line analyzed in this study. Generally speaking, the manuscript is well described and would be interesting for the academic audience. However, the manuscript requires, from my point of view, changes to improve the quality of the manuscript.

  • Although the authors indicate the RNA-seq data used, I don´t understand why not enlist the genes with the most significant changes. If well the biological process of such genes were provided, I think that will be interesting to know the main genes being affected by H3K79me2, in particular if this epigenetic tag promotes higher transcription levels in the identified genes.
  • I consider it crucial to provide the mRNA expression track of HOX9, 10, and 11 in figure 4 to support the statement of lines 226-27.
  • I would like to know if there is a lack of broad H3K79me2 domain in the HOX9 gene for the other leukemic cells K562 and DND41?
  • It appears there is a contradiction. In lines 156-157, the authors indicate that from their observations, the genes with the broadest H3K79 domain exhibit higher transcript levels but also afirm that a gene having the same features (lines 297-98) did not correlate with higher expression.
  • Finally, the authors should consider improving the discussion section by highlighting the main findings and their biological significance instead to describe them again. 
  • In the conclusion section avoid the discussion since is unnecessary. Instead, the concluding paragraphs should provide something new to the reader. 

Other typos:

  • Lines 51-52: Please, provide a reference.
  • Lines 109-115: Consider summarizing the description of leukemic cells 
  • K563, Loucy, and DND-41. Otherwise, move it to the methods section.
  • Line 259: Please provide some examples of the cell lines: in a variety of mammalian cell lines,…… 

  • Consider creating a sole figure by integrating data from figures 4 and 5.

  • Lines 233-235 of the figure legend must be mentioned in the discussion section. (Through the CXXC domain, KMT2A-AF9 could be 233 recruited to the numerous CpG islands present in this genomic region. KMT-AF9 recruits DOT1L 234 which methylates H3K79). 

Author Response

The revised manuscript is attached.

1. Although the authors indicate the RNA-seq data used, I don´t understand why not enlist the genes with the most significant changes. If well the biological process of such genes were provided, I think that will be interesting to know the main genes being affected by H3K79me2, in particular if this epigenetic tag promotes higher transcription levels in the identified genes.

Response: This manuscript focuses on the effects of H3K79me2 broad domains and their effects on expression of genes involved in leukemia. We have identified H3K79me2 broad domains in several genes in several leukemic cell lines pointing out their role in various leukemias. A study on all genes whose expression is altered by H3K79me2 is beyond the scope of this manuscript.   

  1. I consider it crucial to provide the mRNA expression track of HOX9, 10, and 11 in figure 4 to support the statement of lines 226-27.

Response: We have addressed this comment by adding Figures S2 and S3 in supplementary material (line 224).

  1. I would like to know if there is a lack of broad H3K79me2 domain in the HOX9 gene for the other leukemic cells K562 and DND41?

Response: The answer to this comment is no. This comment was addressed on line 370-371.

  1. It appears there is a contradiction. In lines 156-157, the authors indicate that from their observations, the genes with the broadest H3K79 domain exhibit higher transcript levels but also afirm that a gene having the same features (lines 297-98) did not correlate with higher expression.

Response: This text was deleted and talked about the limitations of the RNA seq data in Discussion (paragraph starting on line 343).

  1. Finally, the authors should consider improving the discussion section by highlighting the main findings and their biological significance instead to describe them again. 

Response: We have revised the Discussion section and highlight the limitations of the RNA Seq and H3K79me2 ChIP seq data sets.

  1. In the conclusion section avoid the discussion since is unnecessary. Instead, the concluding paragraphs should provide something new to the reader. 

Response: We revised the Discussion and add a Conclusion section that provides the path forward in doing the analyses of broad H3K79me2 domains.

Other typos:

  1. Lines 51-52: Please, provide a reference.

Response: The references have been added to lines 51-52.

  1. Lines 109-115: Consider summarizing the description of leukemic cells K563, Loucy, and DND-41. Otherwise, move it to the methods section.

Response: We revised paragraph starting on line 113 to address this comment.

  1. Line 259: Please provide some examples of the cell lines: in a variety of mammalian cell lines,……

Response: On line 296, we added the cells used in the mass spectrometry analyses were THP-1, U937, HL60, and mouse thymus.

  1. Consider creating a sole figure by integrating data from figures 4 and 5.

Response: We have combined the old Figure 4 and 5 to a new Figure 4 (line 241).

  1. Lines 233-235 of the figure legend must be mentioned in the discussion section. (Through the CXXC domain, KMT2A-AF9 could be 233 recruited to the numerous CpG islands present in this genomic region. KMT-AF9 recruits DOT1L 234 which methylates H3K79). 

Response: We have added this sentence to the Discussion on line 373.

Reviewer 4 Report

The present manuscript entitled “Broad H3K79me2 domain identifies critical genes in leukemia” is proposed by Sharma et al for publication in Cells journal and investigates the H3K79me2 landscape in leukemia based on published omics datasets from four cell lines : 1 AML (MOLM-13), 2 T-ALL (Loucy, DND-41) and 1 originated from a blast crisis of CML (K562).

Main points :

Globally, the rational for choosing those cell lines have to be enlightened. The translocations / fusions genes / mutations involved in those cell lines have to be indicated in link with the purpose of this study. The authors used only MOLM-13 cell line to conclude about AML cells. The present analysis is to my point of view preliminary and should be straightened by adding other methylome/transcriptome analyses such as using public data for THP-1, MV4-11, HL-60 and many other AML cell models. Similar observation should be made for the other leukemia sub-types to reinforce the conclusions.

The authors evidenced a link between DOT1L inhibitor treatment and HOXA9 and HOXA cluster expression/histone methylation profiles in MOLM13, a point already known and published.

Tables 1 and 2 do not indicate the normalized Enrichment Score (NES) not the adjusted p-values for each identified GO terms that are necessary to be presented within the full list of top GO terms for each cell lines (or at least the rank with the list of identified GO terms). For this GSEA analyses would also be useful, as well as the presentation of some GSEA enrichment plots and the corresponding heatmaps for genes expression from GO term genes lists.

The sentences “Approximately 60% of the genes with the broad 150 H3K79me2 domain had transcript levels in the top 40% of transcript levels (K562, 61%; Loucy, 61%; DND-41, 56%; MOLM-13, 61%), and about 80% of the genes with the broad H3K79me2 domain being in the top 60% transcript levels (K562, 88%; Loucy, 81%; DND-41, 84%; MOLM-13, 89%). These observations show that the genes with the broadest H3K79me2 domains tend to exhibit higher transcript levels” are too approximative as no lists (that could be in supp data) are presented to comfort the results.

The description of the functional roles of HOXA genes in Table 3 has no real interest in the present manuscript and could be removed to focus on the generated results. Figure 4 : the correspondence between ChIP-seq peaks for DOT1L and H3K79me2 is not clear as the figure covers more that 300000bp. It would be interested to look more precisely at the different HOXA genes.

Others:

Figure 1 should go in supp data

The H3K79me2 ChIP-Seq tracks at JMJD1C gene should be presented too, as it is indicated to be the broadest and analysed on other cell lines datasets.

The list of common/unique genes from Figure 2 should be added in supp data.

In summary, the results are too preliminary to be published in Cells. Bioinformatics analysis needs to be strengthened. Additionally, the authors should validate some of the hypotheses using cell experiments on the same cell models and on leukemia patient samples to assess the conclusions. Otherwise, the manuscript should be proposed in more specialized journals with the pitfall that bioinformatics analysis is in itself classic for publication in journals specific to bioinformatics.

Author Response

The revised manuscript is attached.

1. Globally, the rational for choosing those cell lines have to be enlightened. The translocations / fusions genes / mutations involved in those cell lines have to be indicated in link with the purpose of this study. The authors used only MOLM-13 cell line to conclude about AML cells. The present analysis is to my point of view preliminary and should be straightened by adding other methylome/transcriptome analyses such as using public data for THP-1, MV4-11, HL-60 and many other AML cell models. Similar observation should be made for the other leukemia sub-types to reinforce the conclusions.

Response: We provided a rationale for using the CML, AML and T-cell ALL lines on line 113. We added new information about the MLL-type AML lines MV4-11 and THP-1 on a new section starting on line 263 (“2.3. Broad H3K79me2 domain in AML cell lines”). We found H3K79me2 ChIP Seq data for these lines. We focused on H3K79me2 ChIP Seq data to deliver the revised manuscript in 10 days.

  1. The authors evidenced a link between DOT1L inhibitor treatment and HOXA9 and HOXA cluster expression/histone methylation profiles in MOLM13, a point already known and published.

Response: This is only a very minor point which we adequately cited, and we did not attempt to suggest that this was something that we discovered.

  1. Tables 1 and 2 do not indicate the normalized Enrichment Score (NES) not the adjusted p-values for each identified GO terms that are necessary to be presented within the full list of top GO terms for each cell lines (or at least the rank with the list of identified GO terms). For this GSEA analyses would also be useful, as well as the presentation of some GSEA enrichment plots and the corresponding heatmaps for genes expression from GO term genes lists.

Response: We have revised the Tables that show only GO term (biological processes) using Enrichr. This program delivers a rich analysis to populate the table which we hope addresses to some extent this comment.

  1. The sentences “Approximately 60% of the genes with the broad 150 H3K79me2 domain had transcript levels in the top 40% of transcript levels (K562, 61%; Loucy, 61%; DND-41, 56%; MOLM-13, 61%), and about 80% of the genes with the broad H3K79me2 domain being in the top 60% transcript levels (K562, 88%; Loucy, 81%; DND-41, 84%; MOLM-13, 89%). These observations show that the genes with the broadest H3K79me2 domains tend to exhibit higher transcript levels” are too approximative as no lists (that could be in supp data) are presented to comfort the results.

Response: We addressed the limitations of the RNA Seq data in the Discussion (lines 343-352). We now show transcript tracks for the HOXA9, 10 and 11 genes in Figure S2. Excel files with the top 5% of the broadest H3K79me2 domains and RNA Seq data are attached as Supplementary files.

  1. The description of the functional roles of HOXA genes in Table 3 has no real interest in the present manuscript and could be removed to focus on the generated results.

Response: We have moved this Table into supplementary material, and it is now Table S2.

  1. Figure 4 : the correspondence between ChIP-seq peaks for DOT1L and H3K79me2 is not clear as the figure covers more that 300000bp. It would be interested to look more precisely at the different HOXA genes.

Response: We have added a Figure S2 which shows H3K79me2 ChIP-Seq tracks for HOXA9, HOXA10 and HOXA11 genes in MOLM-13 cells.

Others:

  1. Figure 1 should go in supp data

Response: The authors prefer to keep this figure in the main text.

  1. The H3K79me2 ChIP-Seq tracks at JMJD1C gene should be presented too, as it is indicated to be the broadest and analysed on other cell lines datasets.

Response: In Figure S1 we show the H3K79me2 ChIP-Seq track from JMJD1C gene in MOLM-13 cells

  1. The list of common/unique genes from Figure 2 should be added in supp data.

Response: The list of common/unique genes from Figure 2 is added to the supplementary excel file (Tables S1 and S3).

  1. In summary, the results are too preliminary to be published in Cells. Bioinformatics analysis needs to be strengthened. Additionally, the authors should validate some of the hypotheses using cell experiments on the same cell models and on leukemia patient samples to assess the conclusions. Otherwise, the manuscript should be proposed in more specialized journals with the pitfall that bioinformatics analysis is in itself classic for publication in journals specific to bioinformatics.

Response: Our study focuses on analysis of broad domains in leukemic cell lines. Human tumor analysis like the kind suggested by the reviewer is beyond the scope of this study

Round 2

Reviewer 4 Report

1. Globally, the rational for choosing those cell lines have to be enlightened. The translocations / fusions genes / mutations involved in those cell lines have to be indicated in link with the purpose of this study. The authors used only MOLM-13 cell line to conclude about AML cells. The present analysis is to my point of view preliminary and should be straightened by adding other methylome/transcriptome analyses such as using public data for THP-1, MV4-11, HL-60 and many other AML cell models. Similar observation should be made for the other leukemia sub-types to reinforce the conclusions.

Response: We provided a rationale for using the CML, AML and T-cell ALL lines on line 113. We added new information about the MLL-type AML lines MV4-11 and THP-1 on a new section starting on line 263 (“2.3. Broad H3K79me2 domain in AML cell lines”). We found H3K79me2 ChIP Seq data for these lines. We focused on H3K79me2 ChIP Seq data to deliver the revised manuscript in 10 days.

 New request: The authors should estimate the time to perform this analysis and ask the editor for a delay. As the conclusion are at the moment only based on ONE AML cell line, namely MOLM13, the analyses on at least two other cell lines is mandatory unless the manuscript should be rejected.

 10. In summary, the results are too preliminary to be published in Cells. Bioinformatics analysis needs to be strengthened. Additionally, the authors should validate some of the hypotheses using cell experiments on the same cell models and on leukemia patient samples to assess the conclusions. Otherwise, the manuscript should be proposed in more specialized journals with the pitfall that bioinformatics analysis is in itself classic for publication in journals specific to bioinformatics.

 Response: Our study focuses on analysis of broad domains in leukemic cell lines. Human tumor analysis like the kind suggested by the reviewer is beyond the scope of this study

New request : In its present form and if authors do not plan to reinforce the manuscript as requested in point 1 or in point 10, the manuscript does not match with the standards for publication in Cells and should be published in a journal that is more specialized in bioinformatic analysis and I do not doubt that they would also request to comfort the conclusion by analyzing data from additional AML cell lines.

 Other requests are correctly answered. 

Author Response

Comment: New request: The authors should estimate the time to perform this analysis and ask the editor for a delay. As the conclusion are at the moment only based on ONE AML cell line, namely MOLM13, the analyses on at least two other cell lines is mandatory unless the manuscript should be rejected.

Response to point 1: In the revised manuscript, we included two other AML cells lines (MV4-11 and THP-1). It is unclear why the reviewer did not note this.

Comment 2: New request : In its present form and if authors do not plan to reinforce the manuscript as requested in point 1 or in point 10, the manuscript does not match with the standards for publication in Cells and should be published in a journal that is more specialized in bioinformatic analysis and I do not doubt that they would also request to comfort the conclusion by analyzing data from additional AML cell lines.

Response to comment 2: Again we point out that we have analyzed three AML cell lines in the revised manuscript. We also improved the analyses of the GO terms in providing four new tables.